# ID AND OOD PERFORMANCE ARE SOMETIMES INVERSELY CORRELATED ON REAL-WORLD DATASETS

## ABSTRACT

**Context.** Several studies have empirically compared in-distribution (ID) and out-of-distribution (OOD) performance of various models. They report frequent positive correlations on benchmarks in computer vision and NLP. Surprisingly, they never observe *inverse* correlations suggesting necessary trade-offs. This matters to determine whether ID performance can serve as a proxy for OOD generalization.

**Findings.** This paper shows that inverse correlations between ID and OOD performance do happen in real-world benchmarks. They could be missed in past studies because of a biased selection of models. We show an example on the WILDS-Camelyon17 dataset, using models from multiple training epochs and random seeds. Our observations are particularly striking with models trained with a regularizer that diversifies the solutions to the ERM objective (Teney et al., 2022a).

**Implications.** We nuance recommendations and conclusions made in past studies.

- High OOD performance may sometimes require trading off ID performance.
- Focusing on ID performance alone may not lead to optimal OOD performance: it can lead to diminishing and eventually negative returns in OOD performance.
- Our example reminds that empirical studies only chart regimes achievable with existing methods: care is warranted in deriving prescriptive recommendations.

## 1 INTRODUCTION

**Past observations.** This paper complements existing studies that empirically compare in-distribution (ID) and out-of-distribution[1] (OOD) performance of deep learning models (Andreassen et al., 2021; Djolonga et al., 2021; Miller et al., 2021; Mania & Sra, 2020; Miller et al., 2020; Taori et al., 2020; Wenzel et al., 2022). It has long been known that models applied to OOD data suffer a drop in performance, e.g. in classification accuracy. The above studies show that, despite this gap, **ID and OOD performance are often positively correlated**[2] across models on benchmarks in computer vision (Miller et al., 2021) and NLP (Miller et al., 2020).

**Past explanations.** Frequent positive correlations are surprising because nothing forbids opposite, inverse ones. Indeed, ID and OOD data contain different associations between labels and features. One could imagine e.g. that an image background is associated with class $\mathcal{C}_1$ ID and class $\mathcal{C}_2$ OOD. The more a model relies on the presence of this background, the better its ID performance but the worse its OOD performance, resulting in an inverse correlation. Never observing inverse correlations has been explained with the possibility that **real-world benchmarks might contain**

---

[1] We use "OOD" to refer to test data conforming to covariate shifts (Shimodaira, 2000) w.r.t. training data.
[2] We use "correlation" to refer both to linear and non-linear relationships.

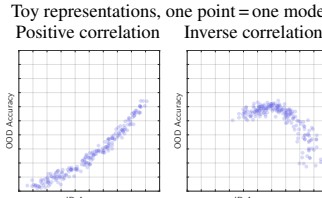

Figure 1: Past studies suggest that positive correlations between ID/OOD performance are ubiquitous. This paper shows, with a counterexample, that inverse correlations are possible and can be accidentally overlooked. The possible need for an ID/OOD trade-off is thus not merely theoretical and should be envisioned, e.g. preventing blind reliance on ID performance for model selection.

**only mild distribution shifts** (Mania & Sra, 2020). We will show that such observations can also be an artefact of study design.

**A recent large-scale study.** Wenzel et al. (2022) show that not all datasets display a clear positive correlation. The authors observe other patterns that sometimes reveal underspecification (D'Amour et al., 2020; Teney et al., 2022b; Lee et al., 2022), or severe shifts that prevent any training/test transfer. Surprisingly, they never observe inverse correlations:

> "*We did not observe any trade-off between accuracy and robustness, where more accurate models would overfit to spurious features that do not generalize.*" (Wenzel et al., 2022)

On the contrary, we do observe such cases and showcase it on a dataset from the above study.

**Explaining inverse correlations.** We name the underlying cause a *misspecification*, by extension of *underspecification* which was previously used to explain why models with similar ID performance can vary in OOD performance (D'Amour et al., 2020; Teney et al., 2022b; Lee et al., 2022). In cases of misspecification, the standard ERM objective (empirical risk minimization), which drives ID performance, conflicts with the goal of OOD performance. ID and OOD metrics can then vary independently and inversely to one another. In Section 5, we present a minimal theoretical example that illustrate how an inverse correlation pattern originates from the presence of both robust and spurious features in the data. In Section 6, we show that different patterns of ID/OOD performance occur with different magnitudes of distribution shifts.

**Summary of contributions.**
- An empirical examination of ID vs. OOD performance on the WILD-Camelyon17 dataset (Koh et al., 2021) that shows an inverse correlation pattern conflicting with past evidence (Section 3).
- An explanation and empirical verification that past studies could miss such patterns because of a biased sampling of models (Section 4).
- A theoretical analysis showing when inverse correlations patterns can occur (Sections 5–6).
- A revision of conclusions and recommendations made in past studies (Section 7).

## 2 PREVIOUSLY-OBSERVED PATTERNS OF ID VS. OOD PERFORMANCE

Past studies conclude that ID and OOD performance tend to vary jointly across models on many real-world datasets (Djolonga et al., 2021; Miller et al., 2021; Taori et al., 2020). Millet al. report an almost-systematic linear correlation[3] between probit-scaled ID and OOD accuracies. Mania & Sra (2020) explain this trend with the fact that real-world benchmarks contain only mild distribution shifts.[4] Andreassen et al. (2021) find that pretrained models perform "above the linear trend" in the early stages of fine-tuning. Their OOD accuracy rises more quickly than their ID accuracy early on, even though the final accuracies agree with a linear trend.

Most recently, the large-scale study of Wenzel et al. (2022) is more nuanced: they observe a linear trend only on some datasets. Their setup consists in fine-tuning an ImageNet-pretrained model on a chosen dataset and evaluating it on matching ID and OOD test sets. They repeat the procedure with a variety of datasets, architectures, and implementation options such as data augmentations.

The scatter plots of ID/OOD accuracy in Wenzel et al. (2022) show four typical patterns (Figure 2).

---

[3]The "*linear trend*" is not really linear: it applies to probit-scaled accuracies (a non-linear transform).

[4]Mania & Sra (2020) explain the linear trend with (1) certain data points having similar probabilities of occurring in ID and OOD data, and (2) the probability being low that a model classifies some points correctly that a higher-accuracy model classifies incorrectly.

Figure 2: Typical patterns observed in Wenzel et al. (2022) (reproduced with permission).

1. **Increasing line (positive correlation): mild distribution shift.** ID and OOD accuracies are positively correlated. Focusing on classical generalization and ID performance brings concurrent OOD improvements.

2. **Vertical line: underspecification** (D'Amour et al., 2020; Lee et al., 2022; Teney et al., 2022b). Different models obtain a similar high ID performance but different OOD performance. The objective of high ID performance does not sufficiently constrains the learning. Typically, multiple features in the data (a.k.a. biased or spurious features) can be used to obtain high ID performance, but not all of them are equally reliable on OOD data. To improve OOD performance, additional task-specific information is necessary, e.g. additional supervision or inductive biases (custom architectures, regularizers, etc.).

3. **Horizontal line, low OOD accuracy: severe distribution shift.** No model performs well OOD. A severe shift prevents any transfer between training and OOD test data. The task needs to be significantly more constrained e.g. with task-specific inductive biases.

4. **No clear trend: underspecification.** Models show a variety of ID and OOD accuracies. The difference with (2) is the wider variety along the ID axis, e.g. because a difficult learning task yields solutions of lower ID accuracy from local minima of the ERM objective.

The authors note the absence of decreasing patterns, which are however possible in theory.

5. **Decreasing line (inverse correlation): misspecification.** The highest accuracy ID and OOD are achieved by different models. Optima of the ERM objective, which are expected to be optima in ID performance, do not correspond to optima in OOD performance. This implies a trade-off: higher OOD performance is possible at the cost of lower ID performance.

---

**When does an inverse correlation occur between ID and OOD performance?**

Intuitively, it can occur when there is a pattern in the data that is predictive in one distribution and misleading in the other. For example, object classes $\mathcal{C}_1$ and $\mathcal{C}_2$ are respectively associated with image backgrounds $\mathcal{B}_1$ and $\mathcal{B}_2$ in ID data, and respectively $\mathcal{B}_2$ in $\mathcal{B}_1$ (swapped) in OOD data. Relying on the background can improve performance on either distribution but not both simultaneously. While such severe shifts might be rare, the next section presents an actual example.

---

## 3 NEW OBSERVATIONS: INVERSELY CORRELATED ID / OOD PERFORMANCE

We use the WILDS-Camelyon17 dataset described by Koh et al. (2021) in a manner similar to Wenzel et al. (2022). These authors evaluate different architectures, assuming different inductive biases will produce models covering a range of ID / OOD accuracies. For simplicity, we rely instead on different random seeds since D'Amour et al. (2020) showed this to be sufficient to cover a variety of ID / OOD accuracies on this dataset. To increase this variety even further without manually picking architectures, we also train models with the diversity-inducing method of Teney et al. (2022a).

**Experimental details.** We use DenseNet-121 models pretrained by the authors of the dataset using 10 different seeds. For each of these 10 models, we re-train the last linear layer from a random initialization for 1 to 10 epochs, while keeping the other layers frozen. These are referred to as "**ERM models**". We perform this re-training with 10 different seeds which gives $10^3$ ERM models (10 pretraining seeds × 10 re-training seeds × 10 numbers of epochs). In addition, we repeat this re-training of the last layer with the diversity-inducing method of Teney et al. (2022a) (details in the box below). These are referred to as "**diverse models**". Each run of the methods produces 24 different models, giving a total of $10^3$. 24 such models ($10^3$ as above × 24).

**Background: learning diverse solutions to a learning task.**

A range of methods exist to identify multiple neural networks of similar high ID performance but that differ in other desirable properties such as OOD performance, interpretability, adversarial robustness, etc. These methods are relevant in cases of underspecification (D'Amour et al., 2020) i.e. when the standard ERM objective does not constrain the solution space to a unique one.

Recent methods consist of optimizing multiple models while encouraging diversity in their **feature space** (Heljakka et al., 2022; Yashima et al., 2022), **prediction space** (Pagliardini et al., 2022; Lee et al., 2022), or **gradient space** (Ross et al., 2018; 2020; Teney et al., 2022a;b).

**This study uses the method of Teney et al. (2022a) which encourages gradient diversity**. The method trains many copies of the same model in parallel – in our case, a linear classifer on top of a frozen DenseNet backbone (see Figure 3).

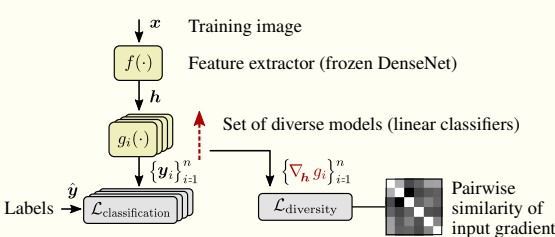

Figure 3: Method used to train a diverse set of models. Each training image $\boldsymbol{x}$ goes through a frozen pre-trained DenseNet to produce features $\boldsymbol{h} = f(\boldsymbol{x})$. We train a set of linear classifiers $\{g_i\}_{i=1}^n$ on these features. A diversity loss minimizes the pairwise similarity between their input gradients.

The models are optimized by standard SGD to minimize the sum of a standard classification loss (cross-entropy) with a diversity loss that encourages diversity across models. Using $\lambda$ a weight hyperparameter, the complete loss is $\mathcal{L} = \mathcal{L}_{\text{classification}} + \lambda \, \mathcal{L}_{\text{diversity}}$. The second term encourages each copy to rely on different features by minimizing the mutual alignment of input gradients:

$$\mathcal{L}_{\text{diversity}} \;=\; \Sigma_{\boldsymbol{x} \in \text{Training}} \; \Sigma_{i=1}^{n} \; \Sigma_{j=i+1}^{n} \; \nabla_{\boldsymbol{h}} \, g_i(\boldsymbol{h}) \, . \, \nabla_{\boldsymbol{h}} \, g_j(\boldsymbol{h}) \quad \text{with} \quad \boldsymbol{h} = f(\boldsymbol{x}) \, . \tag{1}$$

These pairwise dot products quantify the mutual alignment of the gradients. Intuitively, minimizing this loss makes each model locally sensitive along different directions in its input space. Assuming that $g$ produces a vector of logits (as many as there are classes), $\nabla_{\boldsymbol{h}} \, g(\cdot)$ refers to the gradient of the largest logit w.r.t. the classifier's input $\boldsymbol{h}$. We use $n{=}24$ copies and a weight $\lambda{=}10$ that were selected for giving a wide range of ID accuracies. See Teney et al. (2022a) for details about the method. Variations were also described by Ross et al. (2020) and Teney et al. (2022b).

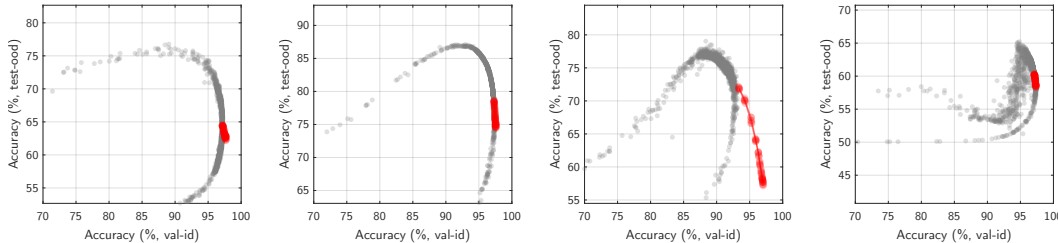

Figure 4: Our new observations show that higher OOD accuracy can sometimes be traded for lower ID accuracy. Each panel corresponds to a different pretraining seed. Each dot represents a linear classifier on frozen features, re-trained with a different seed and/or number of epochs. They are re-trained with standard ERM (red dots ●) or a diversity-inducing method (gray dots ◦). The latter set includes models with higher OOD / lower ID accuracies. See Appendix A for additional plots.

**Results with ERM models.** In Figure 4 we plot the ID vs. OOD accuracy of ERM models as red dots (●). Each panel corresponds to a different pretraining seed. The variation across panels (note the different Y-axis limits) shows that OOD performance varies across pre-training seeds even though the ID accuracy is similar, as noted by Koh et al. (2021). Our new observations are visible *within* each panel. The dots (models) in any panel differ in their re-training seed and/or number of epochs. The seeds induce little variation, but the number of epochs produce patterns of decreasing

trend (negative correlation). Despite the narrow ID variation (X axis), careful inspection confirms that the pattern appears in nearly all cases (see Appendix A for zoomed-in plots).

**Results with diverse models.** We plot models trained with the diversity-inducing method (Teney et al., 2022a) as gray dots (⬤). These models cover a wider range of accuracies and form patterns that extend those of ERM models. The decreasing trend is now obvious. This trend is also clearly juxtaposed with a *rising* trend where ID / OOD performance are positively correlated. This suggests a point of highest OOD performance after which the model overfits to ID data. Appendix A shows similar results with other pretraining seeds. The patterns are not always clearly discernible because large regions of the performance landscape are not covered, despite the diversity-inducing method. We further discuss this issue next.

## 4 WHY PAST STUDIES MISSED NEGATIVE CORRELATIONS: A BIASED SAMPLING OF MODELS

We identified several factors explaining the discrepancy between our observations and past studies.

- ERM models alone do not always form clear patterns (red dots ⬤ in Figure 4). In our observations, the **models trained with a diversity-inducing method** (gray dots ⬤) were key in making the suspected patterns more obvious, because they cover a wider range of accuracies.

- The ID / OOD trade-off varies during training, as noted by Andreassen et al. (2021). This **variation across training epochs** is responsible for much of the newly observed patterns. However, models of different architectures or pretraining seeds are not always comparable with one another because of shifts in their overall performance (see e.g. different Y-axis limits across panels in Figure 4). Therefore the performance across epochs should be analyzed individually per model.

- The "inverse correlation" patterns are not equally visible with all **pretraining seeds**. In some cases, a careful examination of zoomed-in plots is necessary, see Appendix A. This is a reminder that stochastic factors in deep learning can have large effects and that empirical studies should randomize them as much as possible.

To demonstrate these points, we plot our data (same as in Figure 4) while keeping only the ERM models trained for 10 epochs and including all pretraining seeds on the same panel. Figure 5 shows that these small changes reproduce the vertical line observed by Wenzel et al. (2022), which completely misses the inverse correlations patterns visible in Figure 4.

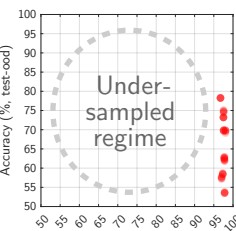

Figure 5: We plot again the ERM models of Figure 4 (red dots ⬤) but **only include models trained for a fixed number of epochs** and combine all pretraining seeds in the same plot. This reproduces the vertical line from (Wenzel et al., 2022), which completely misses the patterns of inverse correlation.

A general explanation is that past studies **undersample regions of the ID / OOD performance space**. They usually consider a variety of architectures in an attempt to sample this space. However, different architectures do not necessarily behave very differently from one another (see the box below). We lack methods to reliably identify models of high OOD performance, but the diversity-inducing method that we use yields models spanning a wide range of the performance spectrum.

---

**Why isn't it sufficient to evaluate a variety of architectures?**

Different architectures do not necessarily induce radically different behaviour. Even CNNs and vision transformers were shown to have similar failure modes (Pinto et al., 2022). Distinct architectures can share similar inductive biases that are e.g. due to SGD, such as the simplicity bias (Scimeca et al., 2022; Shah et al., 2020) or neural anisotropies (Ortiz-Jimenez et al., 2021).

Therefore, independently trained models are not necessarily diverse despite the variety of architectures. ID / OOD performance may only vary along similar directions across models.

---

## 5 THEORETICAL ANALYSIS OF A LINEAR CASE

In this section, we present a minimal case that shows a trade-off between ID and OOD performance and aids understanding the cause of such a pattern. Let $y \in \mathbb{R}$ be a target variable to be predicted by a model, and $\boldsymbol{x}$ the features used as input to the model. These features are a concatenation of invariant and spurious features (defined implicitly below): $\boldsymbol{x} = [\boldsymbol{x}_{\mathrm{inv}} \, ; \, \boldsymbol{x}_{\mathrm{spu}}]$ with $\boldsymbol{x}_{\mathrm{inv}} \in \mathbb{R}^{d_{\mathrm{inv}}}$ and $\boldsymbol{x}_{\mathrm{spu}} \in \mathbb{R}^{d_{\mathrm{spu}}}$. Following Arjovsky et al. (2019); Rosenfeld et al. (2020); Zhou et al. (2022), we consider the simple data-generating process defined by the following structural equations:

$$ y^e \; = \; \boldsymbol{\gamma}^\top \boldsymbol{x}_{\mathrm{inv}}^e + \epsilon_{\mathrm{inv}} \qquad\qquad \boldsymbol{x}_{\mathrm{spu}}^e \; = \; y^e \mathbf{1}^\mathbf{s} + \boldsymbol{\alpha}^e \circ \boldsymbol{\epsilon_{\mathrm{spu}}} \qquad (2) $$

where $e \in \{e_{\mathrm{ID}}, e_{\mathrm{OOD}}\}$ is an environment index referring to ID or OOD data. The random variables $\epsilon_{\mathrm{inv}}$ and $\boldsymbol{\epsilon_{\mathrm{spu}}}$ represent symmetric independent random noise with zero-mean, sub-Gaussian tail probabilities, and $\mathrm{Var}\,(\epsilon_{\mathrm{inv}}) > 0$, $\mathrm{Var}(\epsilon_{\mathrm{spu},i}) > 0$, $\forall\, i \in [1, d_{\mathrm{spu}}]$. The vector $\boldsymbol{\gamma} \in \mathbb{R}^{d_{\mathrm{inv}}}$ determines the relation between the target variable and the invariant features and is identical across environments. In contrast, the vector $\alpha^e$ affects the spurious features and varies among environments. Therefore the *invariant* features are similarly predictive in ID and OOD data while the *spurious* ones are not.

To study the relationship between ID and OOD performance of a hypothetical predictive model, we assume that this model relies on a subset of the features $\boldsymbol{x}$. This subset is identified by a binary mask $\Phi \in \{0,1\}^{d_{\mathrm{inv}} + d_{\mathrm{spu}}}$. Suppose we have already selected $\hat{d}_{\mathrm{inv}}$ invariant features and $\hat{d}_{\mathrm{spu}}$ spurious features, such that $(\hat{d}_{\mathrm{inv}} + \hat{d}_{\mathrm{spu}}) = \hat{d} = ||\Phi_{\hat{d}}||_1$ where the subscript $\hat{d}$ denotes the number of selected features. The features selected by $\Phi_{\hat{d}}$ are $[x_{\mathrm{inv},1}, ..., x_{\mathrm{inv},\hat{d}_{\mathrm{inv}}}, x_{\mathrm{spu},1}, ..., x_{\mathrm{spu},\hat{d}_{\mathrm{spu}}}]$. Let $\mathbb{E}$ denote either the in- and out-of-domain expectation as $\mathbb{E}^{\mathrm{ID}}$ and $\mathbb{E}^{\mathrm{OOD}}$. We use $\beta$ to denote the optimal parameter of the linear regression for a certain domain, i.e., $\beta_{\hat{d}} = \mathbb{E}[\Phi_{\hat{d}}(\boldsymbol{x})^\top \Phi_{\hat{d}}(\boldsymbol{x})]\, \mathbb{E}[\Phi_{\hat{d}}(\boldsymbol{x})^\top y]$. Then the MSE loss of the fitted linear regressor is $\mathbb{E}[y - \Phi_{\hat{d}}(\boldsymbol{x})^\top \beta_{\hat{d}}]^2$. Further, let $[\lambda_1^{\hat{d}}, \lambda_2^{\hat{d}}, ..., \lambda_{\hat{d}}^{\hat{d}}]$ denote the eigenvalues of $\mathbb{E}[\Phi_{\hat{d}}(\boldsymbol{x})^\top \Phi_{\hat{d}}(\boldsymbol{x})]$ and $[\boldsymbol{v}_1^{\hat{d}}, \boldsymbol{v}_2^{\hat{d}}, ..., \boldsymbol{v}_{\hat{d}}^{\hat{d}}]$ the corresponding eigenvectors.

Given a feature mask $\Phi_{\hat{d}}$, we now examine how the ID and OOD losses of the model vary when including an additional spurious feature feature into $\Phi_{\hat{d}}$ (see Appendix B for a proof).

**Theorem 1.** *Including an additional spurious feature leads to the following change in loss $\mathcal{L}$:*

$$ \mathcal{L}_{\mathrm{ID}}(\Phi_{\hat{d}+1}) \quad - \quad \mathcal{L}_{\mathrm{ID}}(\Phi_{\hat{d}}) \quad = \quad \mathbb{E}^{\mathrm{ID}}[y - \Phi_{\hat{d}}(\boldsymbol{x})^\top \beta_{\hat{d}}^{\mathrm{ID}}]^2 \; - \; \mathbb{E}[y - \Phi_{\hat{d}+1}(\boldsymbol{x})^\top \beta_{\hat{d}+1}^{\mathrm{ID}}]^2 \quad < \quad 0 $$

$$ \mathcal{L}_{\mathrm{OOD}}(\Phi_{\hat{d}+1}) - \mathcal{L}_{\mathrm{OOD}}(\Phi_{\hat{d}}) \quad = \quad Q_1 + Q_2 + Q_3 $$

*with $Q_1, Q_2, Q_3$ defined as follows:*

$$ Q_1 \; = \; \mathbb{E}^{\mathrm{OOD}}[y - \Phi_{\hat{d}}(\boldsymbol{x})^\top \beta_{\hat{d}}^{\mathrm{OOD}}]^2 \; - \; \mathbb{E}[y - \Phi_{\hat{d}+1}(\boldsymbol{x})^\top \beta_{\hat{d}+1}^{\mathrm{OOD}}]^2 $$

$$ Q_2 \; = \; \sum_{i=1}^{\hat{d}} \Bigg[ \left(\mathbb{E}^{\mathrm{OOD}}[\Phi_{\hat{d}}(\boldsymbol{x})y]^\top \boldsymbol{v}_i^{\mathrm{OOD},\hat{d}}\right)^2 \left(\lambda_i^{\mathrm{OOD},\hat{d}}\right) \left(\frac{1}{\lambda_i^{\mathrm{ID},\hat{d}}} - \frac{1}{\lambda_i^{\mathrm{OOD},\hat{d}}}\right)^2 $$

$$ - \left(\mathbb{E}^{\mathrm{OOD}}[\Phi_{\hat{d}}(\boldsymbol{x})y]^\top \boldsymbol{v}_i^{\mathrm{OOD},\hat{d}+1}\right)^2 \left(\lambda_i^{\mathrm{OOD},\hat{d}+1}\right) \left(\frac{1}{\lambda_i^{\mathrm{ID},\hat{d}+1}} - \frac{1}{\lambda_i^{\mathrm{OOD},\hat{d}+1}}\right)^2 \Bigg] $$

$$ Q_3 \; = \; \left(\mathbb{E}^{\mathrm{OOD}}[\Phi_{\hat{d}+1}(\boldsymbol{x})y]^\top \boldsymbol{v}_{\hat{d}+1}^{\mathrm{OOD},\hat{d}+1}\right)^2 \frac{((\alpha_{\hat{d}+1}^{\mathrm{ID}})^2 - (\alpha_{\hat{d}+1}^{\mathrm{OOD}})^2)^2}{(\lambda_{\hat{d}+1}^{\mathrm{ID},\hat{d}+1})^2 \, \lambda_{\hat{d}+1}^{\mathrm{OOD},\hat{d}+1}} \quad > \quad 0. $$

*Further, if the new feature is sufficiently unstable in the test domain, i.e. if $((\alpha_{\hat{d}+1}^{\mathrm{ID}})^2 - (\alpha_{\hat{d}+1}^{\mathrm{OOD}})^2)^2$ is sufficiently large such that:*

$$ |(\alpha_{\hat{d}+1}^{\mathrm{ID}})^2 - (\alpha_{\hat{d}+1}^{\mathrm{OOD}})^2| \quad > \quad \sqrt{\frac{(\lambda_{\hat{d}+1}^{\mathrm{ID},\hat{d}+1})^2 \lambda_{\hat{d}+1}^{\mathrm{OOD},\hat{d}+1}}{\left(\mathbb{E}^{\mathrm{OOD}}[\Phi(\boldsymbol{x})y]^\top \boldsymbol{v}_{\hat{d}+1}^{\mathrm{OOD},\hat{d}+1}\right)^2}} \; |Q_1 + Q_2| \,, $$

*then we have $Q_3 > |Q_1 + Q_2|$ and therefore $\mathcal{L}_{\mathrm{OOD}}(\Phi_{\hat{d}+1}) - \mathcal{L}_{\mathrm{OOD}}(\Phi_{\hat{d}}) > 0$.*

Because $\epsilon_{\text{spu}}$ is a zero-mean symmetric random noise, the sign of $\alpha_{\hat{d}+1}$ does not the matter in the results. Theorem 1 shows that adding a spurious feature to those used by the model can affect its ID and OOD losses in opposite directions, implying a trade-off between ID and OOD accuracy. In other words, this minimal case shows that a simple model without/with an extra (spurious) feature can exhibit an inverse correlation between its ID and OOD performance.

## 6  ORDERING ID / OOD PATTERNS ACCORDING TO SHIFT MAGNITUDE

The above analysis shows that inverse correlation patterns are essentially due to the presence of spurious features, i.e. features whose predictive relation with the target in ID data becomes misleading OOD. Occurrences of spurious features increase with the magnitude of the distribution shift. Therefore, the possible patterns in ID / OOD performance presented in Section 2 can be ordered according to the magnitude of the distribution shift they are likely to occur with (see Figure 6).

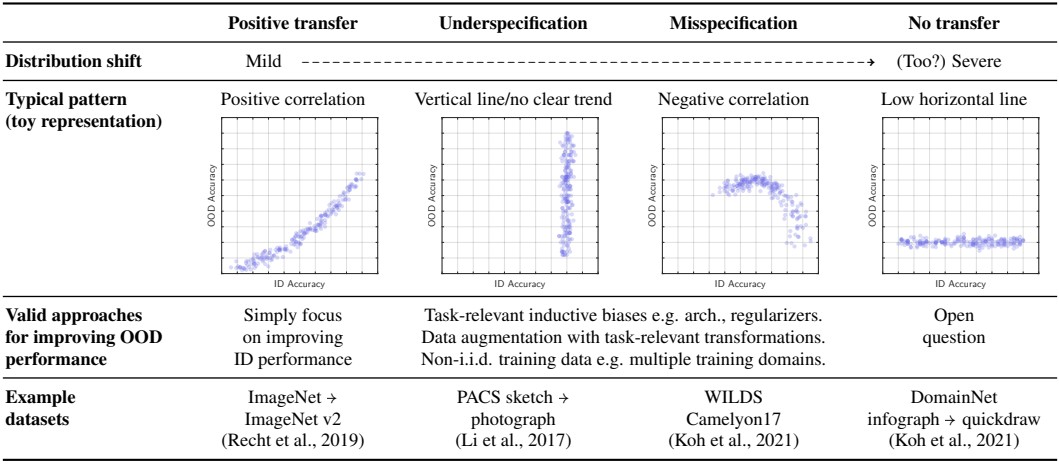

| | Positive transfer | Underspecification | Misspecification | No transfer |
|---|---|---|---|---|
| **Distribution shift** | Mild | - - - - - - - - - - - - - - - - - - - - - - - - - - - - - - - - - → | | (Too?) Severe |
| **Typical pattern (toy representation)** | Positive correlation | Vertical line/no clear trend | Negative correlation | Low horizontal line |
| **Valid approaches for improving OOD performance** | Simply focus on improving ID performance | Task-relevant inductive biases e.g. arch., regularizers. Data augmentation with task-relevant transformations. Non-i.i.d. training data e.g. multiple training domains. | | Open question |
| **Example datasets** | ImageNet → ImageNet v2 (Recht et al., 2019) | PACS sketch → photograph (Li et al., 2017) | WILDS Camelyon17 (Koh et al., 2021) | DomainNet infograph → quickdraw (Koh et al., 2021) |

Figure 6: Various patterns of ID vs. OOD performance occur at different levels of distribution shift.

With the smallest distribution shifts (leftmost case in Figure 6), for example training on ImageNet and testing on its replication ImageNet v2 (Recht et al., 2019), ID validation performance closely correlates with OOD test performance. This OOD setting is the easiest because one can focus on improving classical generalization and reap concurrent improvements OOD.

With a larger distribution shift, more features are likely to be spurious, which is likely to break the ID / OOD correlation. The task of improving OOD performance is likely to be under- or misspecified, i.e. there is not enough information to determine which features a model should rely on to perform well OOD. Valid approaches include modifying the objective function, injecting task-specific information with custom architectures (e.g. building-in invariance to rotations as in Teney & Hebert (2016)), well-chosen data augmentations, or inhomogeneous training data such as multiple training environments (Li et al., 2017) and counterfactual examples (Teney et al., 2020a).

With extreme distribution shifts, most predictive features are overwhelmingly spurious and it is very difficult to learn any one relevant in OOD data (rightmost case in Figure 6).

The proposed ordering of patterns is rather informal and could be further developed following the two axes of diversity shifts and correlation shifts proposed by Ye et al. (2022) (see also Wiles et al. (2021)). More recently, Wang & Veitch (2022) showed that the suitability of various methods for OOD generalization depends on particularities of the underlying causal structure of the task – which must therefore be known to select a suitable method. **Identifying which ID / OOD patterns occur with particular causal structures** might serve as a tool to understand the type of OOD situation one is facing and identify a suitable method.

# 7 REVISITING RECOMMENDATIONS MADE IN PAST STUDIES

We have established that observations in past studies were incomplete. We now bring nuance to some recommendations and conclusions made in these studies.

- **Focusing on a single metric.**

  > "*We see the following potential prescriptive outcomes (...) correlation between OOD and ID performance can simplify model development since **we can focus on a single metric**.*" (Miller et al., 2021)

  We demonstrated that inverse correlations are a possibility, hence there exist scenarios where an ID metric would be misleading. In general, relying on a single metric during model development is ill-advised (Teney et al., 2020b). Even more so here since it cannot capture trade-offs along multiple axes. A model with a suboptimal ID performance may have learned features that enable better OOD generalization. Our recommendation is to track multiple metrics e.g. performance on multiple distributions or qualitative interpretable predictions on representative test points.

- **Improving ID performance for OOD robustness.**

  > "*If practitioners want to make the model more robust on OOD data, **the main focus should be to improve the ID classification error**. (...) We speculate that the risk of overfitting large pretrained models to the downstream test set is minimal, and it seems to be not a good strategy to, e.g., reduce the capacity of the model in the hope of better OOD generalization.*" (Wenzel et al., 2022)

  This recommendation assumes the persistence of a positive correlation. On the opposite, we saw that a positive correlation can precede a regime of inverse correlation (Figure 4, left panels). If the goal is to improve OOD, focusing on ID performance is a blind alley since this goal requires to increase ID performance at times, and reduce it at other times.

- **Future achievable OOD performance.**

  As obvious as it is, it feels necessary to point out that empirical studies only chart regimes achievable with existing methods. Observations have limited predictive power, hence more care seems warranted when deriving prescriptive recommendations from empirical evidence.

  Evidently, the best possible performance on Camelyon17 is not limited to the Pareto front observed in our experiments. For example, the current state of the art on this dataset (Robey et al., 2021; WILDS Leaderboard) injects additional task-relevant knowledge to bypass the under/misspecification of the ERM objective, and exceeds both our highest ID and OOD performance. The important message remains that a given hypothesis class (DenseNet architecture in our case) admits parametrizations whose ID and OOD performance do not necessarily correlate.

- **Possible invalidation of existing studies.**

  The possibility of inverse correlations may invalidate studies that implicitly assume a positive one. For example, Angarano et al. (2022) evaluate the OOD robustness of backbone computer vision architectures. They find that modern architectures surpass domain generalization (DG) methods.

  However, they discard any model but those with the highest ID performance, a.k.a. "*training domain validation*" in Gulrajani & Lopez-Paz (2021). This means that any model with a high OOD performance but non-optimal ID is ignored. They also train every model for a fixed, large number of epochs (30). This may additionally prevent from finding models with high OOD performance since robustness is progressively lost during fine-tuning (Andreassen et al., 2021).

  By design, this study is therefore incapable of finding OOD benefits of any architecture / hyperparameters / DG method that requires trading off some ID performance. Most importantly, once the implicit assumption of a positive correlation is enacted by throwing away models with non-maximal ID performance, there is no more opportunity to demonstrate its validity.

## 8 RELATED WORK

Inverse correlations can be found in the literature, although not always highlighted by their authors.

- An close examination of Gokhale et al. (2022, Table 2) reveals a clear inverse pattern with three benchmarks for natural language inference (NLI). The task is known for biases and shortcuts in the training data, and the OOD test sets in these benchmarks correspond to severe distribution shifts. Our proposed explanation (right end of the spectrum in Figure 6) therefore aligns with these observations. Experiments on question answering from the same authors use data with milder distribution shifts. Correspondingly, they show instead a positive correlation.
- Kaplun et al. (2022, Figure 7) find that the CIFAR dataset contains a subset (CIFAR-10-Neg) on which the performance of visual classifiers is inversely correlated with their ID performance.
- Xie et al. (2020, Section 5.3) discusses cases of inverse correlation on the CelebA dataset with their In-N-Out method – albeit within an overall positive trend.
- McCoy et al. (2019) show that BERT models trained with different random seeds vary widely on their performance on the HANS benchmark for NLI, while their ID performance on the MNLI dataset is similar.
- Naganuma et al. (2022) published an extensive evaluation of OOD benchmarks after the initial release of this paper. They consider a much wider range of hyperparameters than previous studies. As expected from our claims, they observe broader types of relations between ID/OOD performance than the previously reported "linear trend".
- Work on adversarial examples has examined the trade-off between standard and adversarially-robust accuracy Raghunathan et al. (2020); Yang et al. (2020); Zhang et al. (2019). This agrees with our explanation since adversarial inputs correspond to extreme distribution shifts.
- Work on transfer learning has occasionally shown cases of negative transfer, a related phenomenon where improving performance in one domain hurts in others.

## 9 DISCUSSION

This paper showed that inverse correlations between ID / OOD performance are possible not only theoretically, but also happen in real-world data. **We do not know how frequent this situation is.** Although we examined a single counterexample,[5] we also showed that past studies may have systematically overlooked such cases. This suffices to show that one cannot know a priori where a given task falls on the spectrum of Figure 6. It is thus **ill-advised to blindly make the assumption of a positive correlation**, which was suggested in the past.

**Can we avoid inverse correlations with a larger training set?** Scaling alone without data curation seems unlikely to prevent inverse correlations. Fang et al. (2022) examined a more general question and determined that the impressive robustness of the large vision-and-language model CLIP is determined by the *distribution* of its training data rather than its quantity. Similarly, inverse correlations stem from biases in the training distribution (e.g. a class $\mathcal{C}_1$ appearing more frequently with image background $\mathcal{B}_1$ than any other). And biases in a distribution do not vanish with more i.i.d. samples. Indeed, more data can cover more of the support of the distribution, but this coverage will remain uneven, i.e. biased. The problem can become one of "subpopulation shift" (Santurkar et al., 2020) rather than distribution shift, but it remains similarly challenging.

**Training full networks with a diversity-inducing method.** We showed inverse correlations with standard ERM models and – even more strikingly – with linear classifiers trained with a diversity-inducing method (Teney et al., 2022a). To the best of our knowledge, this method has not been applied to deep models because of its computational expense. It would be interesting to confirm our observations on networks trained entirely with this or other diversity-inducing methods (Section 3).

**Qualitative differences along the Pareto frontier.** We focused on quantitative performance. Interpretability methods could also be used to examine whether models of various ID / OOD trade-offs rely on different features and generalization strategies, as done in NLP by Juneja et al. (2022).

Recent work by Eastwood et al. (2022) also recognizes issues of model selection for OOD generalization. They get around selection based on either ID or OOD validation performance with a new

---

[5]We only ran our experiments on Camelyon17. The dataset was not picked post hoc because of unusual results.

domain generalization method (Quantile Risk Minimization) with a tunable trade-off. Other exciting advances by Wang & Veitch (2022) examine existing approaches to improve OOD performance with their suitability to different distribution shifts and underlying causal structures.

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

# A ADDITIONAL RESULTS

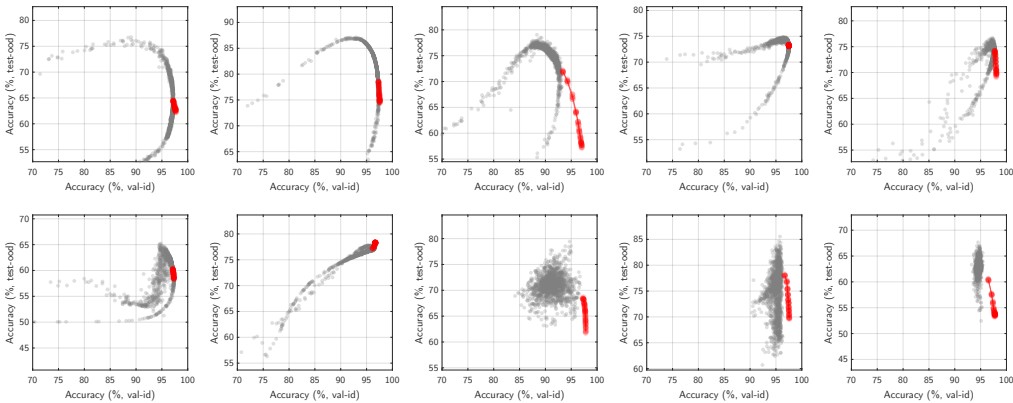

Figure 7: As in Figure 4, we show that higher OOD accuracy can be sometimes be traded off for a lower ID accuracy. Each panel shows results from a different pretrained model (i.e. pretrained with a different random seed). Each dot represents a linear classifier re-trained on features from this pretrained model with standard ERM (red dots ●) or with a diversity-inducing method (Teney et al., 2022a) (gray dots ●). The latter set includes models with higher OOD / lower ID accuracies.

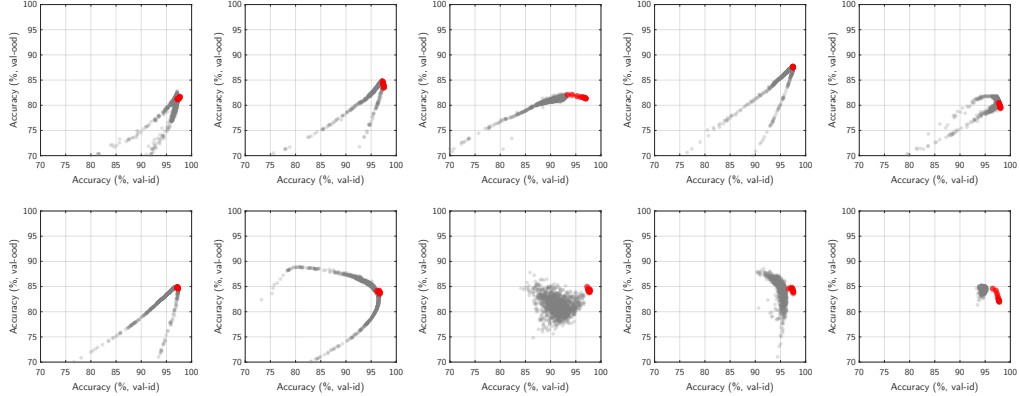

Figure 8: Same as in Figure 7, but using val-ood (instead of test-ood) as the OOD evaluation set.

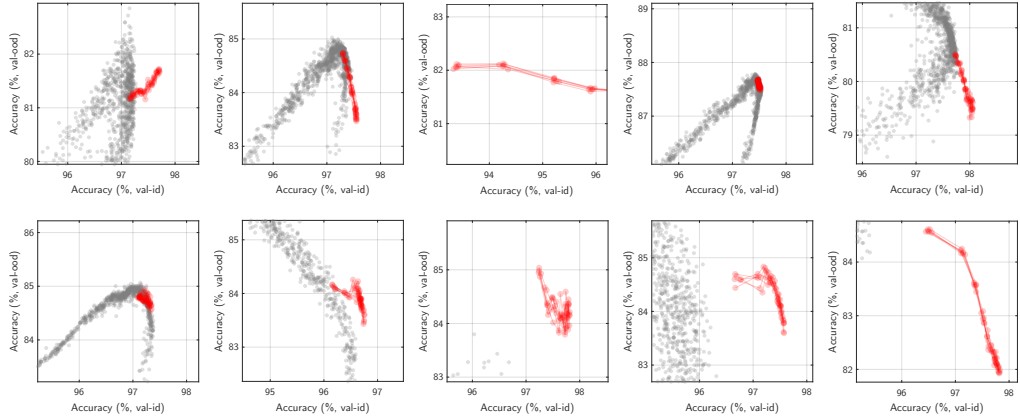

Figure 9: Same as in Figure 8, zoomed-in on ERM models (red dots ●).

# B    PROOF OF THEOREM 1

Let $\boldsymbol{x}_{\hat{d}} := \Phi_{\hat{d}}(\boldsymbol{mx})[\boldsymbol{x}_{\mathrm{inv},1}, ..., \boldsymbol{x}_{\mathrm{inv},\hat{d}_{\mathrm{inv}}}, \boldsymbol{x}_{\mathrm{spu},1}, ..., \boldsymbol{x}_{\mathrm{spu},\hat{d}_{\mathrm{spu}}}]$ be the $\hat{d}$ features already selected and $\boldsymbol{x}_{\hat{d}+1} := \Phi_{\hat{d}+1}(\boldsymbol{mx})$ the features after adding a new spurious feature $\boldsymbol{x}_{\mathrm{spu},\hat{d}_{\mathrm{spu}}+1}$ to $\boldsymbol{x}_{\hat{d}}$.

Let $[\lambda_1^{\hat{d}}, \lambda_2^{\hat{d}}, ..., \lambda_{\hat{d}}^{\hat{d}}]$ denote eigenvalues of $\mathbb{E}[\boldsymbol{x}_{\hat{d}}^{\top}\boldsymbol{x}_{\hat{d}}]$ and $[\boldsymbol{v}_1^{\hat{d}}, \boldsymbol{v}_2^{\hat{d}}, ..., \boldsymbol{v}_{\hat{d}}^{\hat{d}}]$ corresponding eigenvectors.

**Assumption 1.** *The projection of $\mathbb{E}[\boldsymbol{x}_{\hat{d}}^{\top}\boldsymbol{v}_i^{\hat{d}}]$ on each basis corresponding to feature is non zero, i.e.*
$$\left| \mathbb{E}^e[\boldsymbol{x}_{\hat{d}}^{\top}\boldsymbol{v}_i^{\hat{d}}] \right| > 0, \ \ \forall\, e \in \{e_{\mathrm{ID}},\, e_{\mathrm{OOD}}\},\ i \in [d]\,.$$

This ensures that coefficients of a feature can not be always 0, otherwise we can simply remove it.

*Proof.* Let $\beta^{\mathrm{ID}}$ and $\beta^{\mathrm{OOD}}$ denote the solution of linear regression in the ID and OOD domains, i.e.,
$$\beta_{\hat{d}}^{\mathrm{ID}} = \arg\min_{\beta} \mathbb{E}^{\mathrm{ID}}(y - \boldsymbol{x}_{\hat{d}}^{\top}\beta)^2 \tag{3}$$
$$\beta_{\hat{d}}^{\mathrm{OOD}} = \arg\min_{\beta} \mathbb{E}^{\mathrm{OOD}}(y - \boldsymbol{x}_{\hat{d}}^{\top}\beta)^2 \tag{4}$$

Now let us compare the OOD loss after we include $\boldsymbol{x}_{\mathrm{spu},\hat{d}_{\mathrm{spu}}+1}$. In practice, we can only obtain $\beta^{\mathrm{ID}}$ and then apply it on both the ID and OOD domains, which elicits the following errors:
$$\mathcal{L}_{\mathrm{ID}}(\Phi_{\hat{d}}) = \mathbb{E}^{\mathrm{ID}}(y - \boldsymbol{x}_{\hat{d}}^{\top}\beta^{\mathrm{ID}}) \tag{5}$$
$$\mathcal{L}_{\mathrm{OOD}}(\Phi_{\hat{d}}) = \mathbb{E}^{\mathrm{OOD}}(y - \boldsymbol{x}_{\hat{d}}^{\top}\beta^{\mathrm{ID}})$$
$$= \underbrace{\mathbb{E}^{\mathrm{OOD}}(y - \boldsymbol{x}_{\hat{d}}^{\top}\beta^{\mathrm{ID}}) - \mathbb{E}^{\mathrm{OOD}}(y - \boldsymbol{x}_{\hat{d}}^{\top}\beta^{\mathrm{OOD}})}_{\xi_1^{\hat{d}}} + \underbrace{\mathbb{E}^{\mathrm{OOD}}(y - \boldsymbol{x}_{\hat{d}}^{\top}\beta^{\mathrm{OOD}})}_{\xi_2^{\hat{d}}} \tag{6}$$

It is well known that the residual of the linear fitting $y$ by $\boldsymbol{x}_{\hat{d}}$ on the ID domain is
$$\mathcal{L}_{\mathrm{ID}}(\Phi_{\hat{d}}) = \mathbb{E}^{\mathrm{ID}}\big[y - \boldsymbol{x}_{\hat{d}}\mathbb{E}^{\mathrm{ID}}[\boldsymbol{x}_{\hat{d}}^{\top}\boldsymbol{x}_{\hat{d}}]^{-1}\mathbb{E}^{\mathrm{ID}}[\boldsymbol{x}_{\hat{d}}y]\big]^2 = \mathbb{E}^{\mathrm{ID}}[y - \Phi_{\hat{d}}(\boldsymbol{x})^{\top}\beta_{\hat{d}}^{\mathrm{ID}}]^2, \tag{7}$$
Similarly, we have
$$\mathcal{L}_{\mathrm{ID}}(\Phi_{\hat{d}+1}) = \mathbb{E}^{\mathrm{ID}}[y - \boldsymbol{x}_{\hat{d}+1}^{\top}\beta_{\hat{d}+1}^{\mathrm{ID}}]^2. \tag{8}$$

Since $\boldsymbol{x}_{\mathrm{spu},\hat{d}_{\mathrm{spu}}+1}$ does not lies in the space spaned by $\boldsymbol{x}_{\hat{d}}$, so the space spanned by $\boldsymbol{x}_{\hat{d}+1}$ is strictly larger than $\boldsymbol{x}_{\hat{d}}$. Togther with Assumption 1, we have
$$\mathcal{L}_{\mathrm{ID}}(\Phi_{\hat{d}}) - \mathcal{L}_{\mathrm{ID}}(\Phi_{\hat{d}+1}) = \mathbb{E}^{\mathrm{ID}}[y - \boldsymbol{x}_{\hat{d}}^{\top}\beta_{\hat{d}}^{\mathrm{ID}}]^2 - \mathbb{E}^{\mathrm{ID}}[y - \boldsymbol{x}_{\hat{d}+1}^{\top}\beta_{\hat{d}+1}^{\mathrm{ID}}]^2 > 0, \tag{9}$$
and also
$$\xi_2^{\hat{d}} - \xi_2^{\hat{d}+1} = \mathbb{E}^{\mathrm{OOD}}[y - \boldsymbol{x}_{\hat{d}}^{\top}\beta_{\hat{d}}^{\mathrm{OOD}}]^2 - \mathbb{E}^{\mathrm{OOD}}[y - \boldsymbol{x}_{\hat{d}+1}^{\top}\beta_{\hat{d}+1}^{\mathrm{OOD}}]^2 > 0. \tag{10}$$

By the proof in Appendix B.6.3 (above Eq. 29) in Zhou et al. (2022), we have
$$\xi_1^{\hat{d}} = \sum_i^{\hat{d}} (\mathbb{E}^{\mathrm{OOD}}[\boldsymbol{x}_{\hat{d}}^{\top}y]^{\top}\boldsymbol{v}_i^{\mathrm{OOD},\hat{d}})^2 \lambda_i^{\mathrm{OOD}} \Big(\frac{1}{\lambda_i^{\mathrm{IID},\hat{d}}} - \frac{1}{\lambda_i^{\mathrm{OOD},\hat{d}}}\Big)^2. \tag{11}$$

By Eq. (20) in Zhou et al. (2022), we have
$$\lambda_i^{\mathrm{IID},\hat{d}} - \lambda_i^{\mathrm{OOD},\hat{d}} = (\alpha_i^{\mathrm{IID}})^2 - (\alpha_i^{\mathrm{OOD}})^2. \tag{12}$$

So we have:
$$\xi_1^{\hat{d}+1} - \xi_1^{\hat{d}+1} = \sum_{i=1}^{\hat{d}} \Bigg[ \big(\mathbb{E}^{\mathrm{OOD}}[\boldsymbol{x}_{\hat{d}}y]^{\top}\boldsymbol{v}_i^{\mathrm{OOD},\hat{d}}\big)^2 \big(\lambda_i^{\mathrm{OOD},\hat{d}}\big) \Big(\frac{1}{\lambda_i^{\mathrm{ID},\hat{d}}} - \frac{1}{\lambda_i^{\mathrm{OOD},\hat{d}}}\Big)^2$$
$$- \big(\mathbb{E}^{\mathrm{OOD}}[\boldsymbol{x}_{\hat{d}+1}y]^{\top}\boldsymbol{v}_i^{\mathrm{OOD},\hat{d}+1}\big)^2 \big(\lambda_i^{\mathrm{OOD},\hat{d}+1}\big) \Big(\frac{1}{\lambda_i^{\mathrm{ID},\hat{d}+1}} - \frac{1}{\lambda_i^{\mathrm{OOD},\hat{d}+1}}\Big)^2 \Bigg]$$
$$+ \big(\mathbb{E}^{\mathrm{OOD}}[\boldsymbol{x}_{\hat{d}+1}y]^{\top}\boldsymbol{v}_{\hat{d}+1}^{\mathrm{OOD},\hat{d}+1}\big)^2 \frac{((\alpha_{\hat{d}+1}^{\mathrm{ID}})^2 - (\alpha_{\hat{d}+1}^{\mathrm{OOD}})^2)^2}{(\lambda_{\hat{d}+1}^{\mathrm{ID},\hat{d}+1})^2\, \lambda_{\hat{d}+1}^{\mathrm{OOD},\hat{d}+1}}. \tag{13}$$

From Eq. 11 and 13, we have the desired result. $\qquad\square$

