# OpenReview forum: "ID and OOD Performance Are Sometimes Inversely Correlated on Real-world Datasets"
_ICLR.cc/2023/Conference — Submitted to ICLR 2023_

### Official Review · Reviewer_HGmQ · 2022-10-20

**Confidence:** 3
**Correctness:** 2
**Technical Novelty And Significance:** 2
**Empirical Novelty And Significance:** 3
**Recommendation:** 3

**Clarity, Quality, Novelty And Reproducibility:**

The clarity and quality of the experimental part are good.  I think the experiments are reproducible. Novelty is limited. The quality of the recommendations and conclusions needs to be improved.

**Strength And Weaknesses:**

Strength:
1. This work verifies the trade-off between ID and OOD performance happening in a real-world dataset.
2. The authors present detailed discussions about the experimental details and the comparison to the past studies.
3. The authors provide a theoretical analysis of a linear model.

Weakness:
(I list my questions in this section.)

1. The authors do not clearly introduce the OOD generalization task and evaluation.
    - **Q1.** Is it a generalization problem from the training dataset to a specific test dataset? or from the training dataset to a set of multiple test datasets?
    - **Q2.** Do you consider the average accuracy or the worst-case (e.g. worst-domain) accuracy?

2. The authors do not discuss the assumptions or conditions of the OOD generalization problem.
    - **Q3.** For example, if there is no additional information about the unseen test datasets and only the training dataset is accessible, what are the reasons to reject a model with good ID performance? Is this a subjective guess on test data, rather than a scientific model selection scheme?
    - **Q4.** Do the negative correlations between ID and OOD performance imply that the OOD generalization task is unlearnable?

3. This work focuses on a specific case of OOD generalization from the perspective of models and algorithms, but ignores the weakness of the common-used datasets, e.g. the domain generalization benchmarks.
    - **Q5.** We consider a multidomain dataset, e.g. PACS. According to the negative correlation between ID and OOD accuracy, is it possible that there exists a fifth domain such that the current SOTA domain generalization method performs worse than ERM?
    - **Q6.** Can we conclude that current domain generalization benchmarks lack enough domains to evaluate OOD performance?


**Summary Of The Paper:**

This work discovers a concrete example on the WILDS-Camelyon17 dataset that ID performance is negatively correlated to OOD performance. A theoretical example based on a simple linear model is also presented to show the trade-off between ID and OOD performance. The authors explain why past studies missed such a negative correlation and bring recommendations to the OOD generalization studies.

**Summary Of The Review:**

See Strength and Weakness.

---

> ### Author Response · Authors · 2022-11-06
> **Response to Reviewer HGmQ**
>
> Thanks for the precise questions!
>
> > Q1. Is it a generalization problem from the training dataset to a specific test dataset? or from the training dataset to a set of multiple test datasets?
>
> We simply have a training set and two (ID/OOD) test sets (as in most of the related literature discussed in the introduction). The distribution shift incurred in the OOD test set depends on the dataset. The standard assumption is that of a covariate shift. Concretely, with Camelyon17, the OOD data is collected in different hospitals.
>
> &nbsp;
> > Q2. Do you consider the average accuracy or the worst-case (e.g. worst-domain) accuracy?
>
> N/A, no "domains" are used.
>
> &nbsp;
> > Q3. For example, if there is no additional information about the unseen test datasets and only the training dataset is accessible, what are the reasons to reject a model with good ID performance? Is this a subjective guess on test data, rather than a scientific model selection scheme?
>
> In this case (no information about test data) no rational choice can be made either way. As you know, handling a distribution shift fundamentally requires information/assumptions about the shift (i.e. what is invariant). Hence our main message is that the assumption of a positive correlation is not universally valid, and thus cannot be made blindly.
>
> &nbsp;
> > Q4. Do the negative correlations between ID and OOD performance imply that the OOD generalization task is unlearnable?
>
> Obviously, if nothing is known about the shift, it is unlearnable (i.e. anything could shift between training and test time). It is learnable only if assumptions/information about the shift/test distribution is available.
>
> &nbsp;
> > Q5. We consider a multidomain dataset, e.g. PACS. According to the negative correlation between ID and OOD accuracy, is it possible that there exists a fifth domain such that the current SOTA domain generalization method performs worse than ERM?
>
> Yes, this is a corollary of the fact stated above. If there is no assumption/restriction on the distribution shift, for any model, one can construct test data where the model performs very poorly (by introducing a shift on whatever feature the model assumes to be invariant).
>
> Even DG methods (using multiple training domains) require additional (untestable) assumptions about the data-generating process. Hence there exist "fifth domains" that violate whatever assumption a given method makes.
>
> The crux of solving/understanding DG is to determine assumptions powerful enough to enable OOD generalization across useful/realistic shifts, that is not violated by "natural data" (even though it could still be violated by pathological worst-case data).
>
> &nbsp;
> > Q6. Can we conclude that current domain generalization benchmarks lack enough domains to evaluate OOD performance?
>
> Yes and no! More domains would facilitate DG because fewer assumptions would be needed. But it is also possible that there exist assumptions powerful enough to sufficiently constrain the learning task of existing DG benchmarks.
>
>
> Do you think some of these clarifications should be added to the paper? Also let us know if you'd like supporting references for them.

---

### Official Review · Reviewer_yZ1A · 2022-10-24

**Confidence:** 4
**Clarity, Quality, Novelty And Reproducibility:** See above
**Correctness:** 2
**Technical Novelty And Significance:** 3
**Empirical Novelty And Significance:** 3
**Recommendation:** 3

**Strength And Weaknesses:**

I appreciate the thorough exposition on prior works and past observations in the paper’s introduction. Experimentally, I think the use of diversity-inducing methods to enlarge the sampling space of models is clever and well motivated. The recommendation and discussion sections are also particularly informative, highlighting possible over-simplifications in conclusions drawn by prior works.

However, I am leaning towards recommending rejection of this work in its current state. There are three important weaknesses in this paper that limits its contribution towards our collective knowledge about OOD generalization.

The first weakness is that the empirical observations made in this work are entirely based on experiments conducted on the Camelyon17-Wilds dataset. This begs one to question whether the claims made in this paper have any transferability to wider data domains, such as natural images, chest x-rays, satellite images, and NLP datasets. The Camelyon17-Wilds dataset defines “in” and “out” of distribution based on the hospital of origin of the slide image. This is comparatively a narrow scope of ‘in’ distribution, where the distributional shift is almost entirely accounted for by a change in the staining protocol. Other types of distributional shifts, such as those that include both subpopulation shifts and feature changes, might induce different generalization behaviors.

The second weakness lies with the model training protocol. In particular, this work studies the ID and OOD performance of models that arise from training the last layer of a network (a linear model) while varying the initial seed and number of epochs. A danger with this design choice is that observations and theories established for linear models are rarely true for deep neural networks (e.g. the “double descent” generalization curve of over-parameterized models, and the fragility of influence estimation in the deep learning context), and unconverged models behave differently from fully converged models. As such, the observed negative correlation between ID performance and OOD performance could be the direct result of using gradient descent with linear models. For instance, if one considers a case where some spurious features are weakly and positively correlated with an invariant feature in the training domain, then the predictive power of these spurious features to the label is necessarily smaller than the invariant features. One can then apply standard arguments (e.g. https://www.cs.toronto.edu/~rgrosse/courses/csc2541_2021/slides/lec01.pdf) to show that the implicit regularization effect of the gradient descent algorithm causes the invariant features to be learnt early on, and the spurious (and weakly predictive) features are only learnt later on in the optimization process. If this were true, one would predict that by regularizing the number of epochs in the optimizer, one can achieve better generalization on test domains where the predictive direction of such spurious features are flipped. This seems to be indeed the case as the authors observe that “this variation across training epochs is responsible for much of the newly observed patterns”. Modern deep networks, with both explicit and implicit regularizations, may not require such early-stopping to “unlearn” spurious features, and as such do not behave in the same fashion as the models studied in the paper. The authors argue that diverse architectures are not sufficient when studying these trends, but this doesn’t absolve the authors of the responsibility of investigating architectures beyond linear models if they wish to make any general claim about neural network training.

The third weakness lies in the lack of quantitative evaluations for hyperparameters and choice of metrics. From the plots in Appendix A, one observes that the OOD accuracies vary significantly between different pretraining seeds. Furthermore, not every plot exhibits the negative correlation behavior. A careful investigation of the characteristics of the pretrained feature extractor and visualizing other metrics (such as those used in Wenzel 2022) could hope to elucidate factors that can explain these variations.

Regarding questions, I think most of my concerns can be addressed experimentally through additional datasets and models. I will raise my score accordingly.

Lastly, while not directly affecting my score, I think the presentation of Theorem 1 could be made more concise by replacing several repeated expressions with appropriate symbols. The author might also consider modifying the analysis in replacing the “feature selection” model with a direction of steepest descent argument as I linked above to reduce the amount of simplifications required in the analysis.


**Summary Of The Paper:**

This work studies patterns in the performance of ML models on in-distribution (ID) and out-of-distribution (OOD) data. The central claim of this work is that ID performance can be inversely correlated with OOD performance in practice, which has not been explicitly observed before in deep ML models. This work then provides theoretical justification in an ERM model for why this might occur, and discusses the implications these results have on the broader research field.


**Summary Of The Review:**

See above

---

> ### Author Response · Authors · 2022-11-06
> **Response to Reviewer yZ1A**
>
> Thanks for the thorough review and insightful comments.
>
> > W1: Camelyon17-Wilds has a narrow narrow scope of ‘in’ distribution (...)  transferability to wider data domains?
>
> Agreed, any dataset has its idiosyncrasies, obviously. We propose to clarify in the paper that the value is in showing that a "worst case" (inverse correlation) is a possibility to be considered for any future dataset (not just pathological toy data). In other words, the premise of a positive correlation is not a valid universal assumption (even though this is routinely done!). Evaluating an arbitrary selection of existing datasets would add little to this message. This is a very different situation from past studies that sought to support the opposite claim that positive correlations are widespread (cf. proof of existence vs. absence).
>
> We also added a new Section 8 with other cases of inverse correlations we identified in the literature, including some originally overlooked by their authors.
>
> The question about types of shifts/generalization behaviors is one of the most interesting to be solved in this line of work; we briefly discuss existing taxonomies of distribution shifts in Sect. 6.
>
> ---
> > W2: training protocol (...) this doesn’t absolve the authors of the responsibility of investigating architectures beyond linear models if they wish to make any general claim about neural network training
>
> Agreed, these are thoughtful remarks, but our claims are not about general neural network training.
>
> We propose to clarify in the paper that the existence of predictors showing an inverse correlation is relevant to the **data and hypothesis space** rather than learning methods. The issue that we point out, is that learning/selection methods may not cover parts of the hypothesis space with high OOD performance. How one (fails to) arrive at these solutions is a very important question, but the understanding of training dynamics with various regularizations and DG objectives is still limited, to our knowledge [1]. Hence the value in demonstrating the phenomenon (theoretically+empirically) on a minimal class (linear models). Optimization methods notwithstanding, higher-capacity models would be less surprising to display such complex performance patterns.
>
> What do you think about adding the mention to the paper that the implicit regularization of GD can indeed explain the shift in features used during optimization? We would also state clearly that the evolution of robustness during training is an important active area of research (e.g. [1,2,3]).
>
> [1] [The Evolution of Out-of-Distribution Robustness Throughout Fine-Tuning](https://openreview.net/forum?id=Qs3EfpieOh)
>
> [2] [Surgical Fine-Tuning Improves Adaptation to Distribution Shifts](https://openreview.net/forum?id=APuPRxjHvZ) (also submitted to ICLR 2023)
>
> [3] [Fine-Tuning can Distort Pretrained Features and Underperform Out-of-Distribution](https://arxiv.org/abs/2202.10054)
>
> ---
> > most of my concerns can be addressed experimentally through additional datasets and models
>
> A large-scale empirical study is beyond our budget. We discussed with Angarano et al. the possibility of re-analyzing their complete results, but it didn't pan out. We already updated the paper with a new Section 8 of other cases from the literature. This includes another extensive evaluation of OOD benchmarks that just came out [4]. It considers a much wider range of hyperparameters than previous ones, and as expected from our claims, they observe broader types of relations between ID/OOD performance than the previously reported "linear trend".
>
> [4] [Empirical Study on Optimizer Selection for OOD Generalization](https://openreview.net/forum?id=i1s663Cqt9), NeurIPS 2022 DistShift Workshop.

---

### Official Review · Reviewer_94SR · 2022-10-25

**Confidence:** 3
**Correctness:** 3
**Technical Novelty And Significance:** 2
**Empirical Novelty And Significance:** 2
**Recommendation:** 3

**Clarity, Quality, Novelty And Reproducibility:**

Related works and background are well written and easy to follow. Other sections can be largely improved with a cleaner and concise organization. The overall novelty is concerning given the current presentation.

**Strength And Weaknesses:**

Strengths:
- Investigation on the impact of spurious features and the correlation between ID and OOD performance is an important task.
- The paper provides extensive background and review of related works.

Weaknesses:
- Organization can be largely improved. For example, Section 1, 2, and 7 are mostly devoted to review of literature and comparison with previous works. A more concise organization would be better. The background section in Section 3 can be moved to Appendix which diverts the attention of readers.
- Insufficient novelty due to lack of in-depth analysis. It is well expected that the presence of spurious features and reliance on it can negatively impact the OOD performance. The analysis only applies to linear models, which is insufficient to support the empirical results.
- Lack of empirical evidence. The paper only uses ERM and regularized ERM as training objectives. No domain generalization methods are investigated, which leads to an open question whether the trend still hold with methods that are specifically designed for improving domain generalization.

**Summary Of The Paper:**

The paper highlights that an inverse correlation can exist between the in-distribution (ID) and out-of-distribution (OOD) performance of a model. Empirically, the paper supports the claim by training different linear heads with ERM and regularized ERM in [1]. The paper also provides a theoretical study for linear models on the negative impact of spurious features.


[1] Teney et al., Evading the simplicity bias: Training a diverse set of models discovers solutions with superior OOD generalization. CVPR 2022

**Summary Of The Review:**

The paper provides an interesting empirical evaluation on a specific phenomenon on the impact of spurious correlation and OOD generalization. However, the paper provides limited theoretical insights and lack sufficient analysis. The empirical study is limited to ERM on a single dataset.

---

> ### Author Response · Authors · 2022-11-06
> **Response to Reviewer 94SR**
>
> > W1: Organization
>
> Thanks for the suggestion. This is an easy fix that we're happy to follow if other reviewers concur. Another option is to clarify which parts are background material and can be skipped, and ensure the paper remains self-contained for all readers. What do you think?
>
> ---
> > W2a: It is well expected that the presence of spurious features and reliance on it can negatively impact the OOD performance
>
> Obviously. But this is not the point of this paper, right?
>
> ---
> > W2b: The analysis only applies to linear models, which is insufficient to support the empirical results.
>
> The value of the analysis is precisely in demonstrating that the phenomenon can occur in a hypothesis space as simple as linear models. High-capacity models would be less surprising in displaying such complex performance patterns.
>
> ---
> > W3: No domain generalization methods are investigated
>
> We propose to clarify in the paper that the existence of predictors showing an inverse correlation is **relevant to the data and hypothesis space** rather than to learning methods.
>
> The issue we point out is that learning methods may not cover parts of the hypothesis space, thus miss predictors of high OOD performance. This is why the experiments use a general-purpose diversity-inducing method, to cover more of the ID/OOD spectrum than specific DG methods (i.e. we avoid the analysis to place restrictions or focus on the specific inductive biases of an arbitrary set of methods).
>
> See also the new Section 8 with other cases of inverse correlations we identified in the literature.

---

### Official Review · Reviewer_nE6K · 2022-10-31

**Confidence:** 4
**Correctness:** 3
**Technical Novelty And Significance:** 2
**Empirical Novelty And Significance:** 2
**Recommendation:** 5

**Clarity, Quality, Novelty And Reproducibility:**

Overall, the evaluation focuses on a single dataset. Using the results from this dataset, in my opinion, the paper attacks a bit of strawman argument. In particular, I do not think many people would disagree with the paper's claim that "Focusing on ID performance alone may not lead to optimal OOD performance". Instead, past work is merely arguing that IID performance can get you surprisingly far.

**Strength And Weaknesses:**

Strengths:
* I like the claim that when trying to study IID-OOD correlations, the space of models that people have considered in past work is actually quite small. And, by training with diversity-promoting regularizers, one can uncover more of the pareto frontier.

Weaknesses:
* I found the paper's central claim to be a bit unsurprising. Concretely, most past work in this area, e.g., the "Accuracy on the line" paper (Miller et al. 2022), claim there is a strong correlation between IID and OOD performance for many datasets and distributions. However, this is not a "for all" claim: the paper is not arguing that every dataset and shift has this same correlation. Instead, they are merely pointing out that this correlation is surprisingly strong and evident in many existing benchmarks.
* For that reason, I found the paper's main result, that one can find a *single* dataset where the correlation does not hold, to be rather unsurprising. Especially given that the dataset they study, WILDS-Camelyon17, is already a dataset for which Miller et al. 2021 says the IID-OOD correlation is rather weak.
* Moreover, there are other cases in the literature where the IID-OOD corelation is weak or breaks down. For example, the NLP spurious correlations body of work points out a lot of cases where similar models generalize in widely different ways (e.g., https://arxiv.org/abs/1911.02969). The CLIP paper presents another case where finetuned models can have different IID-OOD tradeoffs (improving IID can hurt OOD), and there are numerous follow-up work's that try to mitigate this trade-off.

**Summary Of The Paper:**

An emerging observation is the strong empirical correlation between IID and OOD robustness. This paper presents a counterexample, showing that IID-OOD performance can be inversely correlated on the WILDS-Camelyon17 dataset. The paper also presents a series of arguments to counter specific claims presented in past work, bringing nuance to the discussion of IID-OOD correlation.

**Summary Of The Review:**

The evaluation is quite limited, and the paper is more of a position piece. Overall, I do not find the argument extremely compelling.

---

> ### Author Response · Authors · 2022-11-06
> **Response to Reviewer nE6K**
>
> Agreed with the reviewer. There's however a message that may not be as obvious to all readers.
>
>
> > W1: not many people would disagree that "Focusing on ID performance alone may not lead to optimal OOD performance"
>
> We also would like this to be self-evident! Yet, recent papers have precisely concluded that focusing exclusively on ID performance was a fine strategy:
>
> - "*If practitioners want to make the model more robust on OOD data, the main focus should be to improve the ID classification error*" (Wenzel et al.)
>
> - "*We see the following potential prescriptive outcomes: (...) the correlation between OOD and ID performance can simplify model development since we can focus on a single metric*" (Millet et al.)
>
> What do you think about these? We think that many readers would benefit from our discussion of the necessary caveats to these statements (cf. the effect in subsequent work in response to W2 below).
>
> ---
>
> > W2: most past work (...) is not a "for all" claim (...) merely pointing out that this correlation is surprisingly strong in many benchmarks
>
> Agreed, this is indeed what the experiments show e.g. in "*accuracy on the line*". However:
>
> - The take-away in several papers is not as equivocal. This is evident in the recommendations (cited above) and in subsequent work that takes these observations too far and use the phenomenon as an **unverified assumption** on other datasets. When this assumption serves to justify an experimental design (e.g. model selection) there may be no more chance to verify its validity/recover from a faulty choice, because it was taken as a premise (as already discussed in Section 7; maybe we should expand?). The risk of such methodological mistakes in future research is why we care about this paper.
>
> - The paper also shows how past studies might have over-estimated of the prevalence of these correlations. A large-scale re-evaluation would be nice, but it's not even necessary to show that a "worst case" (inverse correlation) is a possibility for any new dataset (not just pathological toy data). An evaluation of an arbitrary selection of existing datasets would add little to this message.
>
> &nbsp;
> A large-scale empirical study is anyway beyond our resources. We discussed with Angarano et al. the possibility of re-analyzing their complete results but it didn't pan out. However we updated the paper (Sect. 8) with other cases we found in the literature, some of which were originally overlooked by their authors. The given reference is highly relevant but mostly shows underspecification (i.e. weak ID/OOD correlation).
>
> Another extensive evaluation of OOD benchmarks just came out [1]. It considers a much wider range of hyperparameters than previous studies. As expected from our claims, they also observe broader types of relations between ID/OOD performance than the previously reported "linear trend".
>
> [1] [Empirical Study on Optimizer Selection for OOD Generalization](https://openreview.net/forum?id=i1s663Cqt9), NeurIPS 2022 DistShift Workshop.

---

### Decision · Program_Chairs · 2023-01-20

**Decision:**

Reject

**Justification For Why Not Higher Score:**

The main weakness of the paper, in the AC's opinion, are

"I found the paper's central claim to be a bit unsurprising."

"The first weakness is that the empirical observations made in this work are entirely based on experiments conducted on the Camelyon17-Wilds dataset."

I think it's perhaps unfortunate timing for this paper. The AC has to agree with the reviewer that these days many papers have shown that there could be tradeoffs between ID and OOD performance (e.g., the model soup paper, LP-FT paper, etc.), and it becomes much less surprising that ID and OOD can have a negative correlation. If this paper showed up 2 or 3 years ago when those papers on the strong correlation between imagenet and imagent 2.0 results were just out soon, I felt that ut would be much more timely.

**Justification For Why Not Lower Score:**

N/A

**Metareview: Summary, Strengths And Weaknesses:**

The main weakness of the paper, in the AC's opinion, are

"I found the paper's central claim to be a bit unsurprising."

"The first weakness is that the empirical observations made in this work are entirely based on experiments conducted on the Camelyon17-Wilds dataset."

I think it's perhaps unfortunate timing for this paper. The AC has to agree with the reviewer that these days many papers have shown that there could be tradeoffs between ID and OOD performance (e.g., the model soup paper, LP-FT paper, etc.), and it becomes much less surprising that ID and OOD can have a negative correlation. If this paper showed up 2 or 3 years ago when those papers on the strong correlation between imagenet and imagent 2.0 results were just out soon, I felt that ut would be much more timely.